# Hierarchical Granularity Transfer Learning

**Shaobo Min[1], Hongtao Xie[1], Hantao Yao[2],\* Xuran Deng[1], Zheng-Jun Zha[1], Yongdong Zhang[1]**

[1]University of Science and Technology of China, Hefei, China

[2] Institute of Automation, Chinese Academy of Sciences. Beijing, China

mbobo@mail.ustc.edu.cn, {htxie,zhazj,zhyd73}@ustc.edu.cn, hantao.yao@nlpr.ia.ac.cn

## Abstract

In the real world, object categories usually have a hierarchical granularity tree. Nowadays, most researchers focus on recognizing categories in a specific granularity, *e.g.,* basic-level or sub(ordinate)-level. Compared with basic-level categories, the sub-level categories provide more valuable information, but its training annotations are harder to acquire. Therefore, an attractive problem is how to transfer the knowledge learned from basic-level annotations to sub-level recognition. In this paper, we introduce a new task, named Hierarchical Granularity Transfer Learning (HGTL), to recognize sub-level categories with basic-level annotations and semantic descriptions for hierarchical categories. Different from other recognition tasks, HGTL has a serious granularity gap, *i.e.,* the two granularities share an image space but have different category domains, which impede the knowledge transfer. To this end, we propose a novel Bi-granularity Semantic Preserving Network (BigSPN) to bridge the granularity gap for robust knowledge transfer. Explicitly, BigSPN constructs specific visual encoders for different granularities, which are aligned with a shared semantic interpreter via a novel subordinate entropy loss. Experiments on three benchmarks with hierarchical granularities show that BigSPN is an effective framework for Hierarchical Granularity Transfer Learning.

## 1   Introduction

In the real world, object categories usually form a hierarchical tree of different granularities [5, 33, 21, 48], *e.g.,* a hierarchical tree of bird is shown in Fig. 1. For example, a bird has a basic-category "Albatross" and several sub(ordinate)-categories, such as "Footed Albatross" and "Sooty Albatross" species. Compared with basic-level categories, the sub-level categories contain more information, but the annotations are also harder to obtain [41], which require expert taxonomy knowledge to distinguish subtle differences. Thus, how to recognize sub-categories without sub-level image annotations is an interesting and important problem.

To address this issue, we introduce a new task of Hierarchical Granularity Transfer Learning (HGTL), which targets to recognize the subordinate-level categories with only basic-level image annotations and semantic descriptions for hierarchical categories, *e.g.,* attributes [39], as shown in Fig. 1. The insight of HGTL is inspired by the semantic cognition of human. For example, when informed that the "Footed Albatross" has brown wing and the "Sooty Albatross" has black wing, human can distinguish these two sub-species visually.

Among the existing visual recognition tasks, the fine-grained visual categorization (FGVC) [16, 17, 49, 24], domain adaptation (DA) [27–29], and zero-shot learning (ZSL) [12, 13, 39, 23] are most related to HGTL. Different from FGVC that depends on sub-level image annotations, HGTL requires only basic-level annotations and extra semantic information. Compared with DA whose categories of two domains are overlapped, HGTL has disjoint category domains, *i.e.,* basic-categories and

---

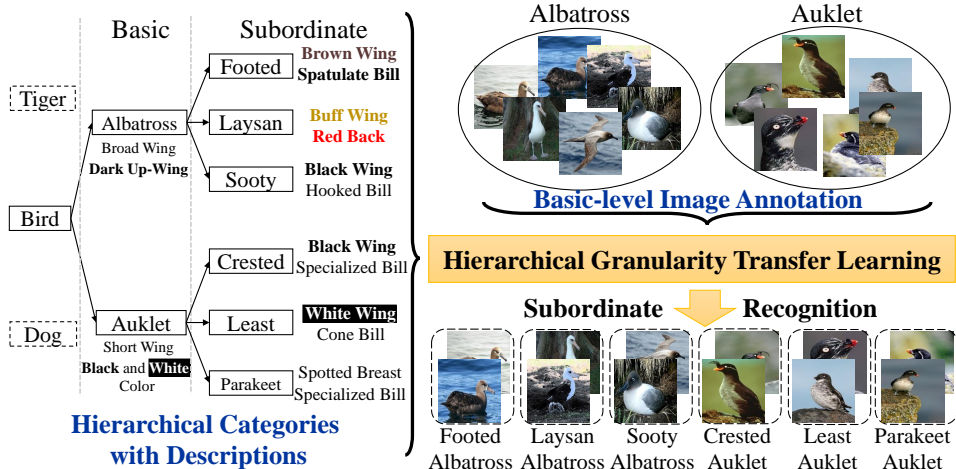

Figure 1: An example of Hierarchical Granularity Transfer Learning (HGTL). Given the basic-level image annotations and category descriptions for hierarchical categories, HGTL aims to recognize the subordinate categories.

sub-categories. ZSL can recognize new categories by transferring the learned visual and semantic embedding functions from seen to unseen domains. However, in HGTL, each image has two disjoint categories, thus a shared visual embedding function cannot well model the visual distributions of two category granularities. In summary, HGTL presents a new challenge of disjoint category domains of two granularities, which has not been explored in the existing recognition methods.

In this paper, we propose a novel Bi-granularity Semantic Preserving Network (BigSPN) to solve the HGTL by constructing two specific visual encoders for respective basic- and sub-domain categories. The core motivation of BigSPN is to leverage the semantic relationship between two category domains for visual knowledge transfer. To this end, BigSPN first learns a visual encoder and a semantic interpreter in the basic domain via the semantic-visual alignment. Since the semantic information can associate two domains, the semantic interpreter is directly transferred to the sub-domain. Then, a new part-based visual encoder is developed to capture the subtle visual difference for sub-category domain. Due to unavailable sub-level image annotations, a subordinate entropy loss is developed to train the new visual encoder to be aligned with the corresponding sub-level semantics, by solving a multi-instance optimization problem. Finally, the sub-domain recognition becomes a nearest neighbor searching problem between part-based visual representations and semantic embeddings for sub-categories. Compared with previous recognition models, BigSPN can preserve the visual distributions for both basic- and sub-domains via two separate visual encoders.

The overall contributions of this paper are summarized by: a) to our best knowledge, we introduce a new task of Hierarchical Granularity Transfer Learning (HGTL) that targets to transfer knowledge between hierarchical categories without subordinate category annotations; b) we propose a novel Bi-granularity Semantic Preserving Network (BigSPN) to bridge the granularity gap for HGTL, by constructing specific visual encoders for hierarchical categories. Due to unavailable sub-level image annotations, the two visual encoders are learned via a shared semantic interpreter and a subordinate entropy loss; c) the evaluations on three benchmarks with hierarchical categories, *i.e.,* CUB-HGTL, AWA2-HGTL, and Flowers-HGTL, demonstrate that the BigSPN is a robust framework for Hierarchical Granularity Transfer Learning.

## 2 Related Work

### 2.1 Fine-Gained Visual Categorization

Different from generic basic-category recognition, fine-grained visual categorization (FGVC) targets to explore the subtle inter-class differences among subordinate object categories [9, 17, 44, 45, 33, 49, 35, 50]. According to [6], FGVC methods can be coarsely categorized into two branches: a)

part-based localization; and b) global-based visual embedding. The part-based methods [9, 32, 49] target to localize important local regions, *e.g,* bird head, for discriminative image representation. Differently, the global embedding methods aim to extract a strong visual representation from a global image directly. However, all the above FGVC methods depend heavily on the image annotations of subordinate categories, which are usually hard to acquire.

## 2.2 Domain Adaptation

Domain adaptation (DA) [27, 29, 18, 43, 46], has been well studied for decades to transfer knowledge between two domains. In DA, the two domains usually have an overlapped label space [3, 19, 30, 46] but different image distributions , *e.g.,* different image styles [27]. To bridge the visual gap between two domains, recent DA methods [20, 29] tend to align the visual representations of two domains so that the source classifier can be directly transferred to the target domain. Different from DA, the proposed HGTL shares an image space but disjoint categories from different granularities, which cannot be solved by using a shared classifier between two domains.

## 2.3 Zero-Shot Learning

Recently, zero-shot learning (ZSL) [2, 13, 15, 39, 42] has attracted increasing attention, which transfers knowledge from the seen categories to the unseen categories. A general paradigm of ZSL is to align the image representations and category descriptions, *e.g.,* category attributes [7, 25] and text descriptions [14], in a joint embedding space. As the semantic information is shared across two domains [7, 8, 39], the learned semantic-visual alignment from the seen domain can be directly transferred to the unseen domain. Under this scheme, recent methods design elaborate visual encoders and semantic interpreters for robust semantic-visual alignment. However, these ZSL methods just consider the disjoint categories in the same granularities, *e.g,* transfer from "Albatross" to "Auklet". When come to different granularities, *e.g.,* transfer from "Albatross" to "Footed Albatross", these methods suffer from a granularity gap, *i.e.,* different granularities share an image space but have different label domains.

# 3 Bi-granularity Semantic Preserving Network

## 3.1 Problem Formulation

The Hierarchical Granularity Transfer Learning (HGTL) targets to recognize sub(ordinate)-level categories with only basic-level annotations and semantic descriptions of hierarchical categories. Formally, we define the image as $I$, the basic-category as $y_b \in \mathcal{Y}_b$, and sub-category as $y_s \in \mathcal{Y}_s$. As shown in Fig. 1, each image $I$ has two categories, which are $y_b$ and $y_s$. $N_b$ and $N_s$ are the class numbers of $\mathcal{Y}_b$ and $\mathcal{Y}_s$. Since $\mathcal{Y}_s$ is the subordinate of $\mathcal{Y}_b$, $N_b \leqslant N_s$. $\boldsymbol{a}(\cdot)$ denotes the semantic descriptions for different categories, such as attributes [7, 25] or text descriptions [14]. Given $\boldsymbol{a}(y_b)$ and $\boldsymbol{a}(y_s)$ along with their affiliation relationship, HGTL targets to train a model using basic-level data pairs $\{I, y_b\}$, that can predict both $y_b$ and $y_s$ for a testing image, and we propose the Bi-granularity Semantic Preserving Network (BigSPN).

## 3.2 Basic-category Recognition

Due to unavailable sub-level annotation $y_s$, BigSPN should first learn from the basic domain data $\{I, y_b\}$, and then leverage the affiliation relationship between $\boldsymbol{a}(y_b)$ and $\boldsymbol{a}(y_s)$ to transfer knowledge to sub domain data. To this end, we first project the images and category descriptions of the basic domain into a joint semantic space by:

$$\mathcal{L}_{sa} = \sum_{\boldsymbol{x}} d[f_v(\boldsymbol{x}), g(\boldsymbol{a}(y_b))] + \mathcal{L}_{cls}(f_v(\boldsymbol{x}), y_b), \qquad (1)$$

where $\boldsymbol{x}$ is the image feature of $I$ generated by backbone network, *e.g.,* ResNet-101. $y_b$ is the basic-annotation, and $\boldsymbol{a}(y_b)$ is the corresponding category description. $f_v(\cdot)$ is a visual encoder to further refine $\boldsymbol{x}$. $g(\cdot)$ is a semantic interpreter that can bridge the semantic-visual gap between $f_v(\boldsymbol{x})$ and $\boldsymbol{a}(y_b)$. $d(\cdot, \cdot)$ is a distance metric function, *e.g.,* the cosine distance [1]. $\mathcal{L}_{cls}(f_v(\boldsymbol{x}), y_b)$ is a

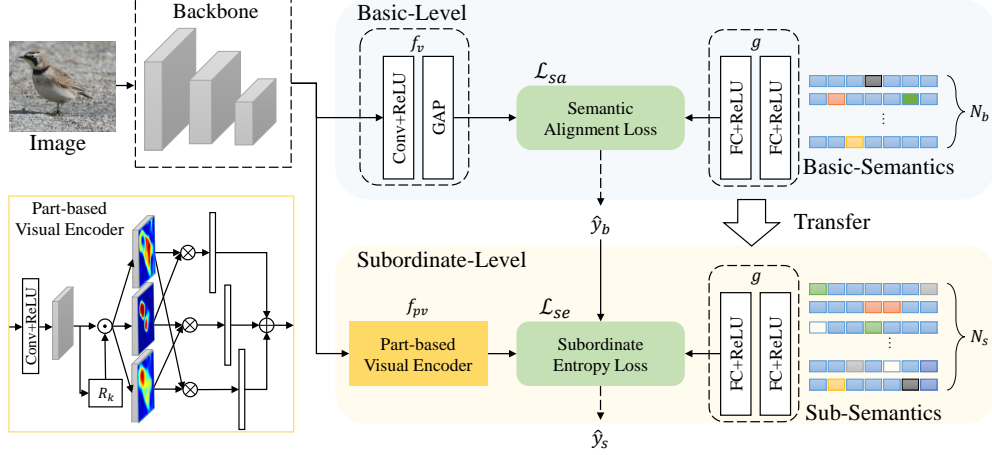

Figure 2: The framework of the proposed Bi-granularity Semantic Preserving Network. GAP indicates global average pooling. The basic-category label of the input image is omitted in $\mathcal{L}_{sa}$. The dashed arrows indicate the inference process, and $\hat{y}_b$ and $\hat{y}_s$ are the outputs of BigSPN for an input image.

standard cross-entropy loss to prevent all $f_v(\boldsymbol{x})$ from being projected into a single point [51], which is defined by $-log\frac{exp(W_{y_b}f_v(\boldsymbol{x}))}{\sum_{y\in\mathcal{Y}_b}exp(W_y f_v(\boldsymbol{x}))}$, where $W_y$ is the classifier weight for class $y$.

By minimizing the semantic alignment loss $\mathcal{L}_{sa}$, the basic-category recognition is converted into a nearest neighbor searching problem by:

$$\hat{y}_b = \arg\min_{y\in\mathcal{Y}_b} d[f_v(\boldsymbol{x}), g(\boldsymbol{a}(y))], \tag{2}$$

where $g(\boldsymbol{a}(y))$ severs as the class anchors, and $f_v(\boldsymbol{x})$ denotes the input queries. The architectures of $f_v(\cdot)$ and $g(\cdot)$ are given in Fig. 2.

### 3.3 Transfer to Subordinate-category Recognition

With the well-trained $f_v(\cdot)$ and $g(\cdot)$, we then explore the affiliation relationship between $\boldsymbol{a}(y_b)$ and $\boldsymbol{a}(y_s)$ to recognize the sub-level categories. Since $\boldsymbol{a}(y_b)$ and $\boldsymbol{a}(y_s)$ share a common semantic space, $g(\cdot)$ can be transferred to $\boldsymbol{a}(y_s)$ to interpret the sub-category descriptions. However, $f_v(\cdot)$ cannot be directly transferred to the sub-domain, because $\mathcal{L}_{sa}$ of Eq. (1) has destroyed the intra-basic-category differences, which is exactly crucial to distinguish its subordinate different categories. Therefore, we target to learn a new visual encoder $f_{pv}(\cdot)$ to specifically recognize sub-categories, by solving two main challenges: a) how to design the architecture of $f_{pv}(\cdot)$ to capture subtle inter-class divergence; and b) without sub-category annotations, how to update the weights of $f_{pv}(\boldsymbol{x})$ to be aligned with correct $\boldsymbol{a}(y_s)$. To address these two issues, we develop a part-based visual encoder $f_{pv}(\cdot)$ and a subordinate entropy loss $\mathcal{L}_{se}$.

#### 3.3.1 Part-based Visual Encoder

To capture the subtle visual clues between sub-level categories, we leverage the multi-attention mechanism to make $f_{pv}(\cdot)$ automatically localize informative regions. The architecture of $f_{pv}(\cdot)$ is given in Fig. 2.

With the backbone image feature $\boldsymbol{x} \in R^{H\times W\times C}$ ($H$, $W$, and $C$ are the height, width, and channel), we first project $\boldsymbol{x}$ into a compressed low dimensional feature $\boldsymbol{z} \in R^{H\times W\times D}$, where $D \ll C$. By taking the feature $\boldsymbol{z}$ as input, we then generate $K$ attentive features by:

$$\boldsymbol{z}_k^{att} = R_k(\boldsymbol{z}) \odot \boldsymbol{z}, k \in [1, K], \tag{3}$$

where $\odot$ denotes the element-wise multiplication. $R_k(\boldsymbol{z}) \in R^{H\times W\times 1}$ is the $k$-th generated attention map, where the value of each pixel indicates the importance of corresponding feature vector in $\boldsymbol{z}$.

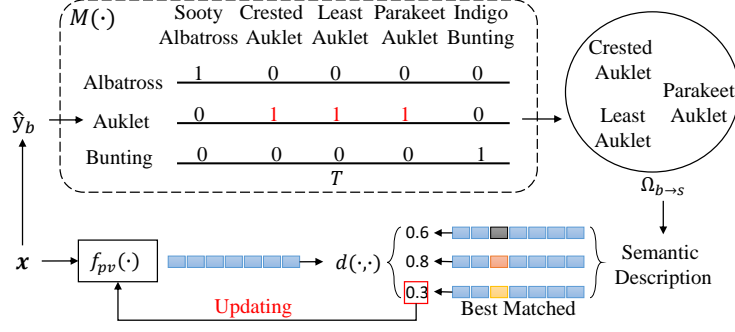

Figure 3: An example of subordinate entropy loss. $\hat{y}_b$ is the predicted basic-category of $x$ using Eq. (2).

$R_k(\cdot)$ is implemented by two convolutions followed by a Sigmoid function. $z_k^{att} \in R^{H \times W \times D}$ is the $k$-th attentive feature for the input $z$. Next, we fuse the attentive features $z_k^{att}(k \in [1, K])$ into a global one, via pairwise bilinear pooling [17]:

$$z^{bp} = \sum_{i=1}^{K-1} \sum_{j=i+1}^{K} z_i^{att} \otimes z_j^{att}, \qquad (4)$$

where $\otimes$ denotes the local pairwise interaction by $a \otimes b = \sum_{n=1}^{N} a_n^{\top} b_n$, where $a_n$ is the feature vector at $n$-th pixel. Finally, $z^{bp} \in R^{D \times D}$ is reshaped to a feature vector by $f_{pv}(x) = vec.(z^{bp})$.

### 3.3.2 Subordinate Entropy Loss

With the sub-level representation $f_{pv}(x)$, a novel subordinate entropy loss $\mathcal{L}_{se}$ is designed to realign $f_{pv}(x)$ with corresponding semantic embedding $a(y_s)$. Notably, the image-level annotations $\{I, y_s\}$ is unavailable during training. The core motivation of $\mathcal{L}_{se}$ is that, although the exact sub-category $y_s$ for $x$ is unknown, we can obtain a sub-category candidate set by using the predicted basic-category $\hat{y}_b$ in Eq. (2).

As $\mathcal{Y}_s$ is the subdivision of $\mathcal{Y}_b$, there is a mapping matrix between them, which is defined by $T \in R^{N_b \times N_s}$. The element at $i$-th row and $j$-th column of $T$ indicates whether the $j$-th sub-category is the subordinate of $i$-th basic-category. With $T$ and predicted $\hat{y}_b$, we can obtain a sub-category candidate set $\Omega_{b \to s}$ by:

$$\Omega_{b \to s} = M(\hat{y}_b, T), \qquad (5)$$

where $M(\cdot)$ is a mapping function that looks up table $T$ in terms of $\hat{y}_b$. A visualized example of $M(\cdot)$ is given in Fig. 3. When $\hat{y}_b$ is predicted correctly, we can be sure that the unavailable $y_s$ of $x$ belongs to $\Omega_{b \to s}$. Thus, we can design $\mathcal{L}_{se}$ by:

$$\mathcal{L}_{se} = \sum_{x} \min_{y \in \Omega_{b \to s}} d[f_{pv}(x), g(a(y))]. \qquad (6)$$

$g(\cdot)$ is fixed which has been well-trained in the basic-domain by Eq. (1).

In Eq. (6), $\mathcal{L}_{se}$ selects the most matched $y$ in $\Omega_{b \to s}$ as the pseudo label to optimize $f_{pv}(x)$. As shown in Fig. 3, $\Omega_{b \to s}$ is a candidate label set that contains the ground truth label of $f_{pv}(x)$. As proved in previous works for multi-instance learning [22, 38, 47], using the most matched instance in the candidate label set can produce a closed-form solution. An intuitive illustration is that, at the beginning of training, $\mathcal{L}_{se}$ constrains $f_{pv}(x)$ to be recognized as one of $\Omega_{b \to s}$. Then, the inherent visual difference among $\Omega_{b \to s}$ makes $f_{pv}(x)$ matched with specific sub-category center $g(a(y))$, where $y \in \Omega_{b \to s}$. As the transferred $g(a(y_s))$ can well describe sub-categories, $f_{pv}(x)$ can be correctly clustered. After minimizing Eq. (6), the matching entropy between $f_{pv}(x)$ and $\Omega_{b \to s}$ is minimized.

Consequently, compared with $f_v(x)$, the semantically realigned $f_{pv}(x)$ can better capture the inter-class differences of sub-categories, resulting in more accurate granularity transfer recognition. The

Table 1: Some statistics of the experimental datasets. *Num.* indicates the category numbers.

| Datasets | Attributes | Basic Num. | Sub Num. | Train | Test |
|---|---|---|---|---|---|
| CUB-HGTL | 312 | 70 | 200 | 5,994 | 5,794 |
| AWA2-HGTL | 85 | 15 | 50 | 22,392 | 14,930 |
| Flower-HGTL | 1024 | 47 | 102 | 4917 | 3272 |

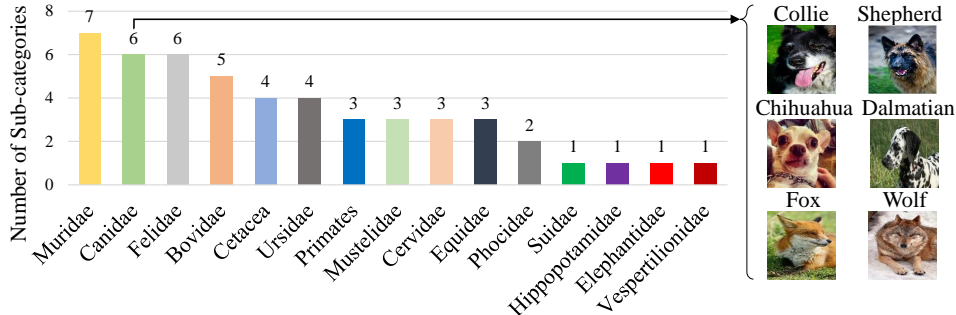

Figure 4: The statistics of hierarchical categories in AWA2-HGTL.

inference function for the sub-category becomes:

$$\hat{y}_s = \arg \min_{y \in \Omega_{b \to s}} d[f_{pv}(\boldsymbol{x}), g(\boldsymbol{a}(y))], \tag{7}$$

where $\Omega_{b \to s}$ is obtained by $M(\hat{y}_b, T)$ in Eq. (5), and the overall objective for BigSPN is:

$$\mathcal{L}_{all} = \mathcal{L}_{sa} + \lambda \mathcal{L}_{se}, \tag{8}$$

where $\lambda$ is a hyper-parameter, and BigSPN is an end-to-end trainable framework.

## 4  Experiments

### 4.1  Experimental Settings

**Datasets.** As the proposed Hierarchical Granularity Transfer Learning (HGTL) is a new task, we construct three datasets with hierarchical categories and semantic descriptions, *i.e.,* CUB-HGTL, AWA2-HGTL, and Flower-HGTL, which are based on the existing datasets of Caltech-USCD Birds-200-2011 [36], Animals with Attributes 2 [39], and Flower [26], respectively.

The CUB contains 200 sub-level bird species along with image-level annotations and category attributes. By clustering the 200 sub-level species based on its specie name, we obtain 70 basic-level categories and the affiliation relationship between two granularities. For each basic-level category, the attribute is generated by averaging its subordinate categories. Finally, we construct the CUB-HGTL dataset whose training set consists of three components: 1) images along with basic-level category annotations; 2) attributes for 70 basic-categories and 200 sub-categories; and 3) affiliation relationship between two category granularities.

In AWA2 and Flower datasets, we construct the hierarchical trees according to biology taxonomy [31, 37] and cluster the sub-level categories of AWA2 and Flower into 15 and 47 basic-level categories, respectively. Similar to CUB-HGTL, the category descriptions for AWA2-HGTL use the attributes [39]. Differently, in Flower-HGTL, the category descriptions [2] use the wiki text, which are embedded into vectors via word2vec. To split the train/val sets for AWA2-HGTL and Flower-HGTL, we randomly divide the images of each sub-category by $3 : 2$, and report the averaged performance for multiple splits. The final data structures of AWA2-HGTL and Flower-HGTL are consistent with CUB-HGTL. The complete category relationship of the three datasets is given in supplementary material.

**Implementation Details.** The backbone network uses the ResNet-101 [11]. MSRA random initializer is used to initialize the BigSPN. In terms of data augmentation, $448 \times 448$ random cropping and horizontal flipping are applied to the input images. Specifically, $\mathcal{L}_{sa}$ and $\mathcal{L}_{se}$ are alternately optimized for each data batch. The batch size is $N = 12$, and reduction channel is $D = 256$.

Table 2: Results of Hierarchical Granularity Transfer Learning on three benchmarks in terms of basic- and subordinate-level categories. R1 and R5 indicate the Rank-1 and Rank-5 accuracy.

| | CUB-HGTL | | | AWA2-HGTL | | | Flowers-HGTL | | |
|---|---|---|---|---|---|---|---|---|---|
| Granularity | Basic_R1 | Sub_R1 | Sub_R5 | Basic_R1 | Sub_R1 | Sub_R5 | Basic_R1 | Sub_R1 | Sub_R5 |
| Domain Adaptation | 92.7 | 24.4 | 57.9 | 98.7 | 35.4 | 88.7 | 89.2 | 35.4 | 66.9 |
| FGN[40] | 93.7 | 26.9 | 60.7 | 94.3 | 45.1 | 90.9 | 86.0 | 36.2 | 65.0 |
| GCNZ[34] | 91.3 | 18.3 | 54.1 | 97.5 | 35.0 | 87.7 | 85.5 | 34.1 | 61.9 |
| SPAEN[4] | 92.7 | 27.0 | 62.9 | 98.5 | 45.2 | 91.1 | 89.1 | 38.5 | 67.7 |
| VSE[51] | 94.1 | 26.0 | 61.5 | 97.7 | 43.3 | 92.7 | 90.3 | 38.5 | 68.1 |
| CosSoftmax[15] | 93.5 | 26.3 | 63.7 | 98.8 | 39.7 | 91.9 | 86.5 | 36.7 | 65.4 |
| BigSPN | 93.3 | **32.8** | **69.9** | 98.3 | **52.0** | **95.4** | 88.7 | **43.0** | **70.9** |

Table 3: The effects of different visual encoders on CUB-HGTL. † indicates that basic visual encoder $f_v$ is directly transferred to the sub-domain, *i.e.,* $f_v$†.

| Methods | | Basic_R1 | Sub_R1 | Sub_R5 |
|---|---|---|---|---|
| $Basic: f_v;$ | $Sub: f_v$† | 92.7 | 24.4 | 57.9 |
| $Basic: f_v;$ | $Sub: f_v$ | 93.1 | 30.1 | 68.1 |
| $Basic: f_v;$ | $Sub: f_{pv}$ | 93.3 | 32.8 | 69.9 |
| $Basic: f_{pv};$ | $Sub: f_{pv}$ | 93.8 | 32.4 | 70.0 |

The SGD optimizer is used with initial $lr = 0.001$, momentum=0.9, and $180$ training epoch. The hyper-parameter is set by $K = 4$ and $\lambda = 1$, which will be analyzed later. During testing, the center part is cropped, and the averaged horizontal flipping results are reported for both basic- and subordinate categories.

**Compared Methods.** As described above, zero-shot learning (ZSL) methods are most related to the HGTL task, thus we mainly compare our BigSPN with six recent ZSL methods: 1) Feature Generation Network (FGN) [40] trains a powerful GAN [10] in the basic space, which can directly generate massive visual representations of sub-categories using the sub category attributes; 2) Graph Convolution Network for ZSL (GCNZ) [34] utilizes the graph convolution architecture to construct $g(\cdot)$, which can better explore the semantic affiliation relationship between the two domains: 3) VSE [51] also explores the local-part embedding to generate discriminative visual representations; and 4) CosSoftmax [15] designs a cosine similarity based Softmax to enhance visual discrimination.

## 4.2 Comparisons

The results on CUB-HGTL, AWA2-HGTL, and Flower-HGTL are summarized in Table 2. In terms of basic-category recognition, all experimental methods obtain comparable results because the inter-class divergences among basic-categories are easy to explore. In terms of sub Rank-1 accuracy, the performance of the compared methods has dropped a lot, *e.g.,* the Sub_R1 is much lower than Basic_R1, which shows that HGTL is a challenging problem. The reason is that, when the single shared $f_v(\cdot)$ of the compared methods is transferred to the sub-space, the minimized basic-category divergence makes the subordinate categories hard to distinguish. Instead, BigSPN constructs two separate visual encoders for basic-level and sub-level categories, which are learned via a shared semantic interpreter and a subordinate entropy loss. Therefore, BigSPN can preserve the inter-class divergence of subordinate categories, and surpasses the compared methods by $5.8\%$, $6.8\%$, and $4.5\%$ for CUB-HGTL, AWA2-HGTL, and Flowers-HGTL, respectively. This proves that the proposed BigSPN is an effective baseline for the new HGTL task.

By evaluating different models on three datasets, we find that the knowledge of basic-categories can be effectively transferred to the sub-categories, with the help of semantic knowledge. For example, BigSPN obtains $52.0\%$ Rank-1 accuracy on AWA2-HGTL dataset. This shows that the HGTL is a feasible task in the real world and has not yet been studied by the existing researchers. In summary, we can conclude that: a) the Hierarchical Granularity Transfer Learning is a feasible, practical, and challenging task; and b) the BigSPN is an effective framework for Hierarchical Granularity Transfer Learning.

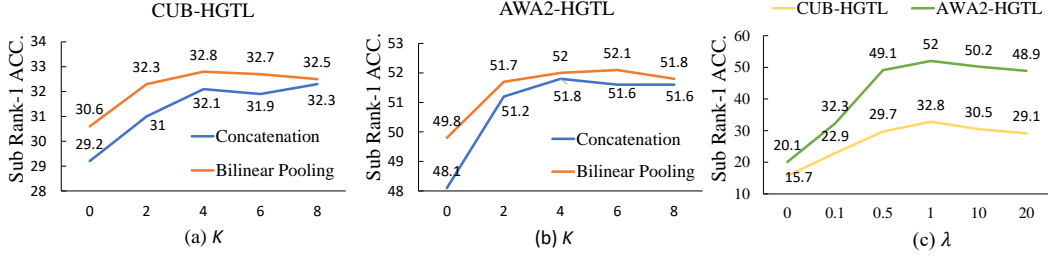

Figure 5: Evaluating the hyper-parameters of $K$ and $\lambda$.

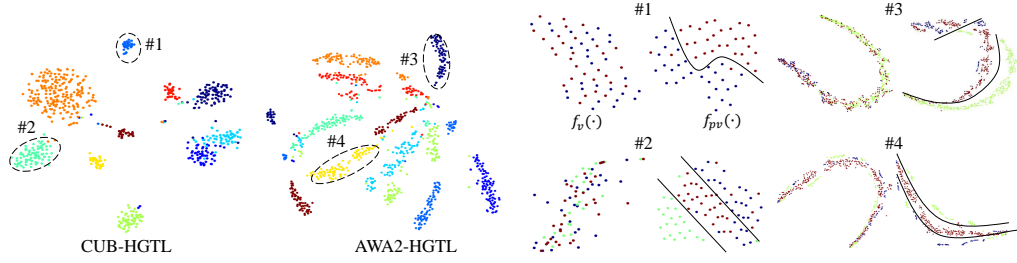

(a) The feature distributions generated by $f_v(\cdot)$.(b) The category subdivision of four clusters in the left Random 10 basic-categories are selected from two figure. The black lines indicate decision boundaries. datasets.

Figure 6: The feature distributions of basic and subordinate categories, that are obtained from $f_v(\cdot)$ and $f_{pv}(\cdot)$ respectively.

## 4.3 Ablation Studies

**Effects of part-based visual encoder.** One of the main differences between BigSPN and related works is the newly learned part-based visual decoder for subordinate categories. We thus analyze its effect and show the results in Table 3, by using different visual encoders for the two category domains. When replacing the transferred $f_v(\cdot)\dagger$ with the newly learned $f_v(\cdot)$, the sub Rank-1 accuracy is improved from $24.4\%$ to $30.1\%$. It proves that there exists a visual distribution difference between two granularities, and using a single shared visual encoder cannot model the granularity gap. Then, we replace the simple 1-layer $f_v(\cdot)$ with the attentive visual encoder $f_{pv}(\cdot)$, and find that the sub Rank-1 accuracy is further improved from $30.1\%$ to $32.8\%$. This shows that the proposed part-based visual decoder $f_{pv}(\cdot)$ can capture more subtle visual clues than the simple $f_v(\cdot)$ for the sub-categories. Finally, when we apply $f_{pv}(\cdot)$ to both domains, no obvious gain is obtained for the sub-domain. In summary, the part-based visual encoder plays a key component in BigSPN.

**Effects of $K$ in $f_{pv}(\cdot)$.** In terms of $K$, we summarize the evaluation results in Fig. 5 (a) and (b). It can be seen that increasing $K$ at the beginning can boost Rank-1 accuracy, and setting $K = 4$ obtains a satisfied result with fewer attentions. Besides, we observe that using the bilinear pooling in Eq. (4) obtains better performance than simply concatenating the features of $z_k^{att}$. Some generated attention map from $R_k(\cdot)$ are visualized in supplementary material.

**Effects of $\lambda$.** Further, we evaluate the effects of $\mathcal{L}_{se}$ in Eq. (8), and report the results in Fig. 5 (c). When $\lambda$ is increased from 0 to 1, the performance is improved obviously. Thus, the subordinate entropy loss $\mathcal{L}_{se}$ can effectively realign the new $f_{pv}(x)$ and corresponding $a_s$, without sub-level annotations. When $\lambda > 10$, the performance drops slightly. Consequently, we find $\lambda = 1$ is suitable for most cases.

**Feature distributions from $f_v(\cdot)$ and $f_{pv}(\cdot)$.** Finally, we give the feature distributions of both basic and subordinate categories that are obtained from $f_v(\cdot)$ and $f_{pv}(\cdot)$. As shown in Fig. 6 (a), the basic-category samples can be well clustered by $f_v(\cdot)$ in two datasets. Moreover, we select four feature clusters of Fig. 6 (a) and further color them according to their subordinate categories in Fig. 6 (b). It can be seen that the features from $f_v(\cdot)$ cannot be separated apart in the sub-category domain. By retraining the new part-based visual encoder $f_{pv}(\cdot)$, the features have much clearer decision

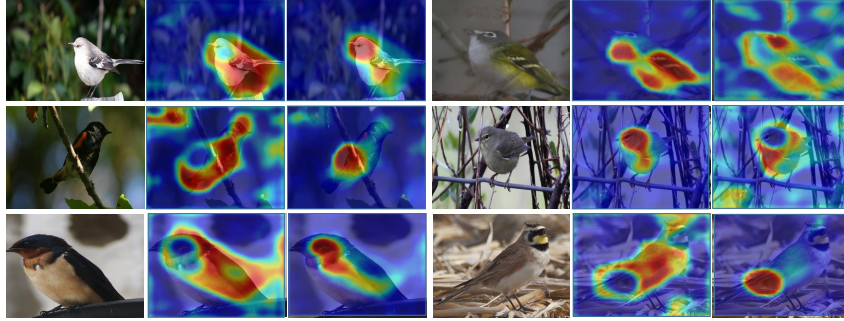

Figure 7: Some results obtained from $f_{pv}(\cdot)$. We randomly select two attention maps generated by $R_k(\cdot)$ for each image.

boundaries than that from $f_v(\cdot)$. This proves the effectiveness of dual visual encoder architecture of BigSPN, as well as the developed part-based visual encoder and subordinate entropy loss.

**Visualized Attention Maps** As illustrated in the main text, the $f_{pv}(\cdot)$ can localize informative local part regions to generate discriminative features. Here, we visualize some generated attention maps for $f_{pv}(\cdot)$ in Figure 7. From the results, by leveraging the attention mechanism, $f_{pv}(\cdot)$ can effectively localize the important regions. Specifically, different attention parts can localize complementary regions, *e.g.,* head and wing, which proved subtle visual clues to distinguish intra-class difference.

## 5    Conclusion

In this paper, we introduce a new task, named Hierarchical Granularity Transfer Learning (HGTL), to recognize the sub(ordinate)-level categories with only basic-level image annotations and extra semantic descriptions of hierarchical categories. Compared with existing tasks, HGTL enables a model to generalize to different granularities without subordinate annotations. Furthermore, we propose a novel framework, named Bi-granularity Semantic Preserving Network, that constructs two separate visual encoders to capture specific distributions for respective basic- and sub-categories. Experiments on three benchmarks prove that the proposed HGTL is a feasible and challenging task, and the BigSPN is an effective framework to transfer knowledge between two granularities.

## 6    Broader Impact

This paper proposes a new visual recognition task, which is general to various recognition scenarios. The positive impacts of this paper contain that: a) the proposed methods enable the data annotators to only label the basic-level images, instead of fine-grained labels, which significantly reduce the annotation difficulty and cost; and b) the proposed model is light and can be easily extended to most existing backbones, which costs little extra computing resource. The negative impacts contain that: a) the proposed HGTL requires abundant semantic annotations for the hierarchical categories, which may be not easy to obtain; and b) the subordinate recognition performance is not so good yet, which should be further improved to apply to practice scenarios.

## 7    Acknowledgements

This work is supported by the National Nature Science Foundation of China (61525206, 62022076, U1936210, 61902399), the National Key Research and Development Program of China (2017YFC0820600), the National Nature Science Foundation of China (61525206, 62022076, U1936210, 61902399), the Youth Innovation Promotion Association Chinese Academy of Sciences (2017209).

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
