[Supplementary Material · sup.pdf]

# Supplementary for Hierarchical Granularity Transfer Learning

## 1 Granularity Gap between Two Domains

Since $\boldsymbol{a}(y_b)$ and $\boldsymbol{a}(y_s)$ share a common semantic space, $g(\cdot)$ can be transferred to $\boldsymbol{a}(y_s)$ to interpret the sub-category descriptions. Thus, an intuitive solution for HGTL is to directly transfer both $f_v(\cdot)$ and $g(\cdot)$ to sub-domain by:

$$\hat{y}_s = \arg\min_{y \in \mathcal{Y}_s} d[f_v(\boldsymbol{x}), g(\boldsymbol{a}(y))]. \tag{1}$$

However, Eq. (1) suffers from the serious granularity gap of visual representations. Specifically, $\mathcal{L}_{sa}$ constrains $f_v(\boldsymbol{x})$ to give a small inner-class divergence of basic-categories, but this inner-class divergence is desired to be preserved to distinguish subordinate categories $y_s$ as shown in Figure 1. Assume that two images $\boldsymbol{x}_1$ and $\boldsymbol{x}_2$ have the same basic-category $y_b$ and different sub-categories $y_s$, minimizing $\mathcal{L}_{sa}$ encourages $f_v(\boldsymbol{x}_1)$ and $f_v(\boldsymbol{x}_2)$ to be aligned with the same $\boldsymbol{a}(y_b)$, thereby making $f_v(\boldsymbol{x}_1)$ and $f_v(\boldsymbol{x}_2)$ hard to separate in the sub-category domain for different $\boldsymbol{a}(y_s)$.

In summary, HGTL suffers from the granularity gap, *i.e.,* some images have the same training basic-category but different testing sub-categories. As shown in Fig. 1 (a), in the basic-domain, we target to minimize the inner-class divergences of "Albatross" and "Auklet" samples to distinguish these two basic-categories accurately. However, when the model is transferred to the sub-domain for testing, all the samples belonging to "Albatross" are clustered compactly, which impedes the further subdivision for "Footed Albatross" and "Laysan Albatross".

## 2 Visualized Attention Maps

As illustrated in the main text, the $f_{pv}(\cdot)$ can localize informative local part regions to generate discriminative features. Here, we visualize some generated attention maps for $f_{pv}(\cdot)$ in Figure 2. From the results, by leveraging the attention mechanism, $f_{pv}(\cdot)$ can effectively localize the important regions. Specifically, different attention parts can localize complementary regions, *e.g.,* head and wing, which proved subtle visual clues to distinguish intra-class difference.

## 3 Some Recognition Results

Some qualitative results of BigSPN for both basic and subordinate categories are given in Fig. 3. It can be seen that the basic-level categories can be easily recognized correctly due to the obvious shape and texture difference. However, when coming to the sub-level categories, some samples are incorrectly recognized.

## 4 Details for datasets

Here, we give detailed hierarchical granularity trees for all three datasets in Table 1, Table 2, and Figure 4. The hierarchical granularity trees are built according to biology Taxonomy [1, 2]. Each basic-level class contains several sub-level categories that have many common characteristics, and

Figure 1: A comparison between general knowledge transfer and the proposed BigSPN.

Figure 2: Some results obtained from $f_{pv}(\cdot)$. We randomly select two attention maps generated by $R_k(\cdot)$ for each image.

different sub-level categories have subtle visual differences. During training stage, CUB-HGTL, AWA2-HGTL, and Flower-HGTL only provide image annotations of basic-categories, and semantic descriptions for both basic and sub(ordinate)-level categories. During testing stage, the task is how to predict the sub-level category for an input image.

The detailed dataset information is given in the "CUB", "AWA2", and "FLO" folders, respectively, as well as the train/val splits and all category descriptions.

# References

[1] Simpson, G.G.: Principles of animal taxonomy (1961)

[2] Wheeler, Q.D.: Taxonomic triage and the poverty of phylogeny. Philosophical Transactions of the Royal Society of London. Series B: Biological Sciences **359**(1444), 571–583 (2004)

Table 1: Detailed hierarchical granularity tree for Flower-HGTL.

| | Basic | Subordinate |
|---|---|---|
| | Ambelliferae | great masterwort. |
| | Aquifoliaceae | alpine sea holly. |
| | Aroideae | anthurium, giant white arum lily. |
| | Asteraceae | artichoke, barbeton daisy, bishop of llandaff, blanket flower, marigold, mexican aster, moon orchid, orange dahlia, osteospermum, oxeye daisy, colt's foot, common dandelion, english marigold, pink-yellow dahlia, purple coneflower, spear thistle, gazania, globe thistle, sunflower. |
| | Ericaceae | azalea. |
| | Campanulaceae | balloon flower, canterbury bells. |
| | Iridaceae | bearded iris, blackberry lily, spring crocus, sword lily, yellow iris. |
| | Lamiaceae | bee balm. |
| | Strelitziaceae | bird of paradise. |
| | Acahaceae | black-eyed susan, mexican petunia. |
| | Nyctaginaceae | bougainvillea. |
| | Bromeliaceae | ball moss, bromelia. |
| | Proteaceae | king protea. |
| | Ranunculaceae | lenten rose, buttercup, clematis, japanese anemone. |
| | Nymphaeaceae | lotus, water lily. |
| | Ranunculaceae | love in the mist, columbine, windflower. |
| | Magnoliaceae | magnolia. |
| | Malvaceae | mallow, hibiscus, tree mallow. |
| Flowers | Monkshood | monkshood. |
| | Convolvulaceae | morning glory, silverbush. |
| | Passifloraceae | passion flower. |
| | Geraniaceae | pelargonium, geranium. |
| | papaveraceae | californian poppy, corn poppy, tree poppy. |
| | theaceae | camellia. |
| | Cannaceae | canna lily. |
| | caryophyllaceae | carnation, sweet william. |
| | Zingiberaceae | cautleya spicata, red ginger. |
| | Primulaceae | cyclamen, pink primrose, primula. |
| | Amaryllidaceae | cape flower, daffodil, peruvian lily, hippeastrum. |
| | Apocynaceae | desert-rose, frangipani. |
| | Solanaceae | petunia, thorn apple. |
| | Euphorbiaceae | poinsettia. |
| | Rosaceae | rose. |
| | Orchidaceae | ruby-lipped cattleya, hard-leaved pocket orchid. |
| | liliaceae | siam tulip, fire lily, fritillary, grape hyacinth, tiger lily, toad lily. |
| | Scrophulariaceae | snapdragon, foxglove. |
| | Polemoniaceae | garden phlox. |
| | onagraceae | gaura. |
| | Tropaeolaceae | globe-flower. |
| | gentianaceae | bolero deep blue, stemless gentian. |
| | Leguminosae | sweet pea. |
| | Bignoniaceae | trumpet creeper. |
| | Cruciferae | wallflower. |
| | Piperaceae | watercress. |
| | Violaceae | wild pansy. |
| | Amaranthaceae | prince of wales feathers. |
| | caprifoliaceae | pincushion flower. |

Table 2: Detailed hierarchical granularity tree for CUB-HGTL.

| | Basic | Subordinate |
|---|---|---|
| Birds | Albatross | Black footed Albatross, Laysan Albatross, Sooty Albatross. |
| | Ani | Groove billed Ani. |
| | Auklet | Crested Auklet, Least Auklet, Parakeet Auklet, Rhinoceros Auklet. |
| | Blackbird | Brewer Blackbird, Red winged Blackbird, Rusty Blackbird, Yellow headed Blackbird. |
| | Bobolink | Bobolink. |
| | Bunting | Indigo Bunting, Lazuli Bunting, Painted Bunting. |
| | Cardinal | Cardinal. |
| | Catbird | Spotted Catbird, Gray Catbird. |
| | Chat | Yellow breasted Chat. |
| | Towhee | Eastern Towhee. |
| | Widow | Chuck will Widow. |
| | Cormorant | Brandt Cormorant, Red faced Cormorant, Pelagic Cormorant. |
| | Cowbird | Bronzed Cowbird, Shiny Cowbird. |
| | Creeper | Brown Creeper. |
| | Crow | American Crow, Fish Crow. |
| | Cuckoo | Black billed Cuckoo, Mangrove Cuckoo, Yellow billed Cuckoo. |
| | Finch | Gray crowned Rosy Finch, Purple Finch. |
| | Flicker | Northern Flicker. |
| | Flycatcher | Acadian Flycatcher, Great Crested Flycatcher, Least Flycatcher, Olive sided Flycatcher, Scissor tailed Flycatcher, Vermilion Flycatcher, Yellow bellied Flycatcher. |
| | Frigatebird | Frigatebird. |
| | Fulmar | Northern Fulmar. |
| | Gadwall | Gadwall. |
| | Goldfinch | American Goldfinch, European Goldfinch. |
| | Grackle | Boat tailed Grackle. |
| | Grebe | Eared Grebe, Horned Grebe, Pied billed Grebe, Western Grebe. |
| | Grosbeak | Blue Grosbeak, Evening Grosbeak, Pine Grosbeak, Rose breasted Grosbeak. |
| | Guillemot | Pigeon Guillemot. |
| | Gull | California Gull, Glaucous winged Gull, Heermann Gull, Herring Gull, Ivory Gull, Ring billed Gull, Slaty backed Gull, Western Gull. |
| | Hummingbird | Anna Hummingbird, Ruby throated Hummingbird, Rufous Hummingbird. |
| | Violetear | Green Violetear. |
| | Jaeger | Long tailed Jaeger, Pomarine Jaeger. |
| | Jay | Blue Jay, Florida Jay, Green Jay. |
| | Junco | Dark eyed Junco. |
| | Kingbird | Tropical Kingbird, Gray Kingbird. |
| | Kingfisher | Belted Kingfisher, Green Kingfisher, Pied Kingfisher, Ringed Kingfisher, White breasted Kingfisher. |
| | Kittiwake | Red legged Kittiwake. |
| | Lark | Horned Lark. |
| | Loon | Pacific Loon. |
| | Mallard | Mallard. |
| | Meadowlark | Western Meadowlark. |
| | Merganser | Hooded Merganser, Red breasted Merganser. |
| | Mockingbird | Mockingbird. |
| | Nighthawk | Nighthawk. |
| | Nutcracker | Clark Nutcracker. |
| | Nuthatch | White breasted Nuthatch. |
| | Oriole | Baltimore Oriole, Hooded Oriole, Orchard Oriole, Scott Oriole. |
| | Ovenbird | Ovenbird. |
| | Pelican | Brown Pelican, White Pelican. |
| | Pewee | Western Wood Pewee. |
| | Sayornis | Sayornis. |
| | Pipit | American Pipit. |
| | Will | Whip poor Will. |
| | Puffin | Horned Puffin. |
| | Raven | Common Raven, White necked Raven. |
| | Redstart | American Redstart. |
| | Geococcyx | Geococcyx. |
| | Shrike | Loggerhead Shrike, Great Grey Shrike. |
| | Sparrow | Baird Sparrow, Black throated Sparrow, Brewer Sparrow, Chipping Sparrow, Clay colored Sparrow, House Sparrow, Field Sparrow, Fox Sparrow, Grasshopper Sparrow, Harris Sparrow, Henslow Sparrow, Le Conte Sparrow, Lincoln Sparrow, Nelson Sharp tailed Sparrow, Savannah Sparrow, Seaside Sparrow, Song Sparrow, Tree Sparrow, Vesper Sparrow, White crowned Sparrow, White throated Sparrow. |
| | Starling | Cape Glossy Starling. |
| | Swallow | Bank Swallow, Barn Swallow, Cliff Swallow, Tree Swallow. |
| | Tanager | Scarlet Tanager, Summer Tanager. |
| | Tern | Artic Tern, Black Tern, Caspian Tern, Common Tern, Elegant Tern, Forsters Tern, Least Tern, Green tailed Towhee. |
| | Thrasher | Brown Thrasher, Sage Thrasher. |
| | Vireo | Black capped Vireo, Blue headed Vireo, Philadelphia Vireo, Red eyed Vireo, Warbling Vireo, White eyed Vireo, Yellow throated Vireo. |
| | Warbler | Bay breasted Warbler, Black and white Warbler, Black throated Blue Warbler, Blue winged Warbler, Chestnut sided Warbler, Golden winged Warbler, Hooded Warbler, Kentucky Warbler, Magnolia Warbler, Mourning Warbler, Myrtle Warbler, Nashville Warbler, Orange crowned Warbler, Palm Warbler, Pine Warbler, Prothonotary Warbler, Swainson Warbler, Tennessee Warbler, Wilson Warbler, Worm eating Warbler, Prairie Warbler, Canada Warbler, Cerulean Warbler, Cape May Warbler, Yellow Warbler. |
| | Waterthrush | Northern Waterthrush, Louisiana Waterthrush. |
| | Waxwing | Bohemian Waxwing, Cedar Waxwing. |
| | Woodpecker | American Three toed Woodpecker, Pileated Woodpecker, Red bellied Woodpecker, Red cockaded Woodpecker, Red headed Woodpecker, Downy Woodpecker. |
| | Wren | Bewick Wren, Cactus Wren, Carolina Wren, House Wren, Marsh Wren, Rock Wren, Winter Wren. |
| | Yellowthroat | Common Yellowthroat. |

Figure 3: Some results on CUB-HGTL and AWA2-HGTL. Besides the images, the input of BigSPN includes auxiliary category descriptions for both basic- and sub-levels. The red color indicates incorrect prediction.

Figure 4: Detailed hierarchical granularity tree for AWA2-HGTL.