[Reviews · NeurIPS 2020]

Review 1

Summary and Contributions: This paper presents an interesting problem setting for the domain adaptation field, which focus on both basic-level or sub(ordinate)-level classes. A validate solution, Bi-granularity Semantic Preserving Network, is proposed to address this new setting.

Strengths: The new problem setting is interesting. It is worth to investigate this new setting.

Weaknesses: 1. The new setting is not formalized well from theoretical perspective. 2. Motivations of some components of the proposed method are not clear. 3. There are no theoretical guarantees for the proposed method.

Correctness: Yes, they are correct.

Clarity: No, many components of the proposed method are not motivated well. I do not understand why the network should be designed like the proposed one.

Relation to Prior Work: Yes, it is clearly discussed how this work differs from previous contributions.

Reproducibility: Yes

Additional Feedback: a) I think that this paper considers an interesting problem. However, this new problem is not formalized well in math, and the solution is not the easy-understanding one. There are many notations and I am not sure what they stand for. b) After we train a DA model, we can just transfer knowledge from the Basic-Semantics to the Sub-Semantics, right? This should be a reasonable baseline rather than directly using DA. c) In the experiment, I notice that DA has a good performance on Basic-Semantics. So, it would be interesting to consider a two-stage solution (mentioned in b)) for this new problem. ------------------------- After Rebuttal Thanks for addressing parts of my concerns! I have carefully read comments from other reviewers and the feedback. I agree with R3 that the technical novelty is limited since the solution to this new problem is an A+B one, which does not contain much novelty. Besides, the new problem is still not formalized well in math in the feedback. The assumptions behind this new problem is unclear. In the feedback, the sentence "Some available side information includes affiliation relationship between..." is unclear. What does these side informance stand for? How to formalize them in math is very important. Since 1) the new setting is not formalized well and 2) the technical novelty is limited (A+B solution), this paper is not good enough to be presented in NeurIPS 2020.


Review 2

Summary and Contributions: This paper introduces a new Hierarchical Granularity Transfer Learning (HGTL) task which targets to recognize sub-categories with only provided basic level annotations and extra semantic descriptions of hierarchical categories. The task is related to the domain adaption and ZSL tasks, but it is much harder due to the granularity gap between two different granularities. Thus, a key problem is how to preserve different visual distributions for a sample when transferring knowledge between granularities. To this end, they propose a novel Bi-granularity Semantic Preserving Network (BigSPN), which follows ZSL by aligning the visual features with corresponding category descriptions, while the major difference is that they learn a new Part-based Visual Encoder and a subordinate entropy loss for the sub-level domain.

Strengths: Strengths: The paper is well written and easy to follow. The proposed HGTL task is a critical and attractive problem based on the real vision problem, and it may attract other researchers to follow. The idea of BiSPN is novel and the proposed network is technically sound, which shows better performance than the state-of-the-art DA and ZSL methods on the proposed HGTL task. Extensive evaluations on a few visual datasets are conducted to prove the methods.

Weaknesses: Weakness: - I think there should be a connection paragraph between section 3.2 and 3.3 to illustrate why a new visual encoder is needed. - A baseline model, e.g., ZSL method, should be given to explain how the related ZSL methods are applied to the proposed HGTL task. - Attention map part needs more explanation - Why does BigSPN perform not well on Basic_R1 as compare to other state-of-the-art ZSL methods? Any discussion? - How do you determine the epochs =180, learning rate =0.001 and batch size=12? How about the learning rate scheduler? - More visualized recognition examples should be given.

Correctness: Yes, the claim of this paper is clear, and the empirical methodology is correct

Clarity: Yes. This paper proposes a new task for granularity transfer learning, as well as a baseline solution, which is clearly clarified. The paper is well-written and easy to follow.

Relation to Prior Work: Yes. There are three types of related works mentioned in this paper. The difference is that FGVC requires sub-level annotations, DA assumes shared label space across two domains, and ZSL cannot handle different granularities. Thus, I think the difference from previous works is clear and the contribution is sufficient.

Reproducibility: Yes

Additional Feedback: [After rebuttal] I think this paper presents a novel and interesting task, as in industry it is hard to find enough scientists to annotate massive images with fine-grained categories. In terms of the concerns of other Reviewers, I agree that a theoretical formulation for HGTL and BigSPN can make the paper more convincing, but current response and manuscript can explain what HGTL does and how BigSPN solves the task. According to the author response, the categorical hierarchy is indeed available from scientists for limited categories, but the cheap wiki text descriptions for hierarchical categories maybe too noisy to capture enough transferable knowledge, compared to human-annotated category attributes. Using powerful NLP models may alleviate this problem, or not, which should be further explored. As for me, HGTL is a practical task, e.g., automatic pre-annotation for sub-category and model transfer to different scenes. To solve HGTL, BigSPN is simple baseline, which adapts several techniques to HGTL in a new way and obtains good performance. After reading the rebuttal, I keep my original rate.


Review 3

Summary and Contributions: This paper addresses a transfer learning scenario, where the task transferring happens from the basic-level categories and that of sub-categories. The hierarchical category relationships are given, and each category also have a text description associated with it. The paper uses zero-shot learning method and entropy regularization constrained by the category hierarchy and basic-level prediction. The basic-level classifier and the sub-category classifier use different network architecture. Bilinear pooling are used together with multi-head attentive features for sub-category classification. Experiments on done on CUB and AWA2. ---- After rebuttal ----- Thanks for the response. I still feel the work needs more tech strength.

Strengths: The paper proposed an interesting transfer of learning setting. The proposed solution is reasonable in the given problem setting. In particular, it uses a standard way to do learning the image-text joint embedding and put it together with a classification loss. When applying the entropy minimization loss, it simplifies the problem using the hierarchical relationship between the basic-level prediction and the subcategory prediction. The experimental results show that the proposal method works better than purely zero-shot learning methods in the proposed setting.

Weaknesses: The proposed problem setting seems very specific. The setting requires the availability of the categorical hierarchy and the text description of each category, which is a demanding requirement in the real world. It is hard to see a wide range of applications. The proposed method has limited novelty. Technique it used for the joint embedding learning is not new. It is also a straightforward idea to constrain the entropy minimization to the subcategories under the basic-level category. For the subcategory classifier, the usefulness of the multi-head attentive features and bilinear pooling is not clear. According to Figure 5, the benefit of using such a complicated architecture is marginal but requires careful hyper-parameter tuning.

Correctness: Yes. Yes.

Clarity: Yes.

Relation to Prior Work: Yes. In the meanwhile, it may be helpful to make connections with hierarchical classification works.

Reproducibility: Yes

Additional Feedback:


Review 4

Summary and Contributions: The paper proposes a new task named Hierarchical Granularity Transfer Learning (HGTL) and a new network architecture called Bi-granularity Semantic Preserving Network (BigSPN). HGTL has only basic category labels and semantic descriptions for hierarchical categories. The goal is to recognize sub-category levels without annotations for sub-category levels. In this paper, 2 levels (basic, subordinate) are considered. Semantic descriptions are typically attributes, keywords or text descriptions. Knowledge transfer between granularity levels happens via affiliation relationships between attributes across granularities. The key is to align image representations and semantic category descriptions. BigSPN is end-end trainable by alternating optimization of Semantic alignment loss and novel Subordinate entropy loss. The subordinate entropy loss is based on the estimated base category obtained using nearest neighbor search between the part-based visual encoding and the semantic representation. HGTL is differentiated from Fine Grained Visual Categorization (FGVC), Domain Adaptation (DA) and Zero Shot Learning (ZSL). FGVC methods require annotations for subordinates categories. Unlike domain adaptation, HGTL does not use shared classifiers across domains. Advanced ZSL approaches typically align image representations with category descriptions to generalize to unseen categories in the same granularity level. HGTL considers transfer across different granularity levels which share the same image space but not the label space. Semantic-visual alignment is used to learn a visual encoder and semantic interpreter in the basic domain. Part based visual encoder is learned for sub-domains. Semantic interpreter is shared across all domain granularities. Part-based visual encoder extracts representation based on bilinear pooling of attention features generated by modulation of attention map and compressed low dimensional feature. Novel subordinate entropy loss is then used to align this representation with the corresponding semantic embedding. 3 new datasets were created based on existing popular datasets for FGVC, namely CUB-HGTL, AWA2-HGTL, and Flowers-HGTL Experiment section is insightful with comparison of BigSPN with six recent ZSL methods, demonstrating that the BigSPN is a robust framework for Hierarchical Granularity Transfer Learning. Ablation study is comprehensive including 1) effect of part-based visual encoder 2) effects of hyperparameters K, lambda 3) feature distributions ----- After rebuttal: Overall, the novelty gap is small but the problem is of practical importance. Thus downgraded from 9 to 8.

Strengths: The paper proposes a novel task that is very practical, where fine category labels are not provided and need to be recognized using only coarse labels and category level semantic descriptions are available during training. Experiments are solid and include 3 new derived datasets. Relevant work included is sufficient.

Weaknesses: Could have been better if the paper used large datasets such as iNaturalist.

Correctness: Yes. Experiments are sound and convincing.

Clarity: Yes. The paper is well written.

Relation to Prior Work: Yes. This is a novel task and is compared with FGVC, DA and ZSL.

Reproducibility: Yes

Additional Feedback: Incorrect sentence in Figure 6 caption in subfigure.

[Author Response · NeurIPS 2020]

We thank the reviewers for their comments. This work presents a novel HGTL task along with a baseline BigSPN
framework. As agreed by all reviewers, it is an interesting and novel task, which will attract more researches to follow.

**To Review #1:**
**Q1:Unclear formulation of HGTL.** Given the basic-category annotated data $\{\boldsymbol{x}, y_b \in \mathcal{Y}_b\}$ and sub-category set $\mathcal{Y}_s$,
HGTL aims to train a model which minimizes testing recognition error on both $\{\boldsymbol{x}, y_b\}$ and $\{\boldsymbol{x}, y_s \in \mathcal{Y}_s\}$. Some
available side information includes affiliation relationship between $\mathcal{Y}_b$ and $\mathcal{Y}_s$, and category descriptions $\boldsymbol{a}(\cdot)$.
**Q2:A reasonable DA baseline.** Yes, we actually have adjusted DA to HGTL with a two-stage manner. A base DA/ZSL
model first trains a visual encoder $f_v(\cdot)$ and a semantic interpreter $g(\cdot)$ by minimizing $d[f_v(\boldsymbol{x}), g(\boldsymbol{a}(y_b))]$ on $\{\boldsymbol{x}, y_b\}$,
and then directly transfer $f_v(\cdot)$ and $g(\cdot)$ to target domain by argmax $d[f_v(\boldsymbol{x}), g(\boldsymbol{a}(y_s))]$, where $y_s \in \Omega_{b \to s}$. Notably,
$\Omega_{b \to s}$ (line 171) is the subordinate categories of predicted $y_b$ of $\boldsymbol{x}$ by argmax $d[f_v(\boldsymbol{x}), g(\boldsymbol{a}(y_b))]$.
**Q3:Unclear motivation of BigSPN.** Given two images $\boldsymbol{x}_1$ and $\boldsymbol{x}_2$ with the same basic category $y_b$ but different
sub-categories $y_{s1}$ and $y_{s2}$. For $y_b$, we expect $f_v(\boldsymbol{x}_1)$ and $f_v(\boldsymbol{x}_2)$ to be invariant, but for $y_{s1}$ and $y_{s2}$, $f_v(\boldsymbol{x}_1)$ and $f_v(\boldsymbol{x}_2)$
should be discriminative. This conflict motivates us to learn a new visual encoder $f_{pv}(\cdot)$ to specifically recognize $y_{s1}$
and $y_{s2}$. To capture the subtle visual clues between $\boldsymbol{x}_1$ and $\boldsymbol{x}_2$, multi-head attention and bilinear pooling are used
to build $f_{pv}(\cdot)$ in Fig. 2. Due to unavailable annotated $\{(\boldsymbol{x}_1, y_{s1}), (\boldsymbol{x}_2, y_{s2})\}$, we design an entropy minimization
loss $\mathcal{L}_{se}$ in Eq. (7) to train the weights of $f_{pv}(\cdot)$ using only $\{(\boldsymbol{x}_1, \boldsymbol{x}_2, y_b)\}$. The experimental results have proved the
effectiveness of BigSPN in HGTL, which surpasses all related DA&ZSL methods.
**Q4: Two-stage solution for HGTL.** All the compared methods and BigSPN are the two-stage models, as stated in Q2.

**To Review #2:**
**Q1:Illustrating why a new visual encoder is needed.** Please refer to Q3 of R#1.
**Q2:Explaining how ZSL is applied to HGTL.** A baseline ZSL model can refer to Q2 of R#1, which first
trains $\{f_v(\cdot), g(\cdot)\}$ on basic-domain data $\{\boldsymbol{x}, y_b\}$, and then merely transfers $\{f_v(\cdot), g(\cdot)\}$ to $\{\boldsymbol{x}, y_s\}$ by argmax
$d[f_v(\boldsymbol{x}), g(\boldsymbol{a}(y_s))]$. Other related ZSL methods in Table 2 usually design stronger $f_v(\cdot)$ or $g(\cdot)$.
**Q3:Why BigSPN performs not well on Basic_R1?** Since BigSPN focuses on designing a new visual extractor
$f_{pv}(\cdot)$ for the sub-category recognition, we just adopt a simple 1-layer convolution as the visual extractor $f_v(\cdot)$ for
basic-category. Compared to well-designed $f_v(\cdot)$ in ZSL models, e.g., SPAEN [5] uses the auto-encoder architecture,
BigSPN obtains comparable results in Basic_R1, but much better results in Sub_R1.
**Q4:How to determine training strategy.** Grid searching is used to determine the training strategies from the general
training strategies in ZSL and FGVC. Exponential Decay is used with step 30 and decay=0.1.
**Q5:Other issues.** The motivation and design for attention map part please refers to Q3 of R#1. More recognition
examples will be added in revision.

**To Review #3:**
**Q1:applicability in real world about categorical hierarchy and text description.** HGTL focuses on object recog-
nition applications, such as animals, goods, and plants, whose categorical hierarchy naturally exists and is the basic
knowledge for scientists in specific fields. Compared with massive image annotations, the hierarchy for limited
categories is easier for scientists to build. For the text description of categories, it is cheap to collect from Internet,
e.g., querying in wiki, though it maybe noisy. Thus, besides using "word2vec" to encode wiki text for Flower-HGTL,
stronger NLP models, e.g., Bert, will be explored to generate more compact semantic embeddings. With scientist's
categorical hierarchy and category description from wiki-Bert, HGTL can be applied to various real-world applications.
For example, with only basic-category annotations, HGTL can automatically pre-annotate sub-categories to reduce
human labeling burden.
**Q2:Limited technique novelty in BigSPN.** Different from ZSL&DA, each image in HGTL has two categories, as our
answer to Q3 of R#1. Thus, one main contribution of BigSPN is to learn a two-branch architecture for respective basic-
and sub-category recognition without sub-category annotations. Specifically, for sub-category branch, the multi-head
attention $f_{pv}$ is used to capture detailed visual clues, and entropy minimization is leveraged to train the weights of $f_{pv}$
in an unsupervised manner. Although the separate concept of each component is not new, **BigSPN is the first work
that designs specific visual representations for basic- and sub-category recognitions with only basic-category
annotations.** Thus, BigSPN is a novel and effective baseline for the new HGTL task, compared to related ZSL&DA
methods.
**Q3: Use of multi-attention and bilinear pooling.** The motivation of multi-attention and bilinear pooling refers to Q2.
On CUB, they bring obvious 2.7% grain from Table 3, and the tuned hyper-params are suitable for most of datasets.
**Q4: Other issues.** Hierarchical classification requires hierarchical category annotations, which is unavailable in HGTL.
Detailed discussion about hierarchical classification and broader impact will be added in revised version.

**To Review #4:**
**Q1:Extension to larger iNat dataset.** We are building the category descriptions and hierarchy for iNat dataset, and
detailed experiments will be published.
**Q2:Typos.** We have carefully revised the typos and incorrect sentences.

[Meta-Review · NeurIPS 2020]

R1 and R3 comment that the paper lacks mathematical grounding and novelty. However, R2 and R4 both think that the paper proposes an interesting and useful task and could be adopted by vision researchers. I think the paper should be accepted.